# Phosphates Induced H-Type or J-Type Aggregation of Cationic Porphyrins with Varied Side Chains

**DOI:** 10.3390/molecules28104115

**Published:** 2023-05-16

**Authors:** Zhiliang Li, Charles J. Zeman, Silvano Valandro, Jose Paolo O. Bantang, Kirk S. Schanze

**Affiliations:** 1Department of Chemistry, University of Texas at San Antonio, San Antonio, TX 78249, USA; zhiliang.li@sdu.edu.cn (Z.L.); charles.zeman@northwestern.edu (C.J.Z.IV); silvano.valandro@chemie.uni-hamburg.de (S.V.); jopaobantang@gmail.com (J.P.O.B.); 2Department of Chemistry, University of Florida, Gainesville, FL 32611, USA

**Keywords:** aggregation hydrolysis, phosphates, porphyrins, self-assembly

## Abstract

Non-covalent interactions have been extensively used to fabricate nanoscale architectures in supramolecular chemistry. However, the biomimetic self-assembly of diverse nanostructures in aqueous solution with reversibility induced by different important biomolecules remains a challenge. Here, we report the synthesis and aqueous self-assembly of two chiral cationic porphyrins substituted with different types of side chains (branched or linear). Helical H-aggregates are induced by pyrophosphate (PPi) as indicated by circular dichroism (CD) measurement, while J-aggregates are formed with adenosine triphosphate (ATP) for the two porphyrins. By modifying the peripheral side chains from linear to a branched structure, more pronounced H- or J-type aggregation was promoted through the interactions between cationic porphyrins and the biological phosphate ions. Moreover, the phosphate-induced self-assembly of the cationic porphyrins is reversible in the presence of the enzyme alkaline phosphatase (ALP) and repeated addition of phosphates.

## 1. Introduction

A large variety of nanostructures containing self-assembled systems with biological functions have been observed in nature [1,2]. Traditional supramolecular chemistry focusing on non-covalent interactions offers the possibility of realizing a variety of functions in artificial self-assembling systems to mimic biological structures in organic media [3,4,5,6,7]. Porphyrin and perylene derivatives are well known to self-assemble into various supramolecular nanostructures through forming H-type or J-type aggregates depending on the environment and side chain structure [3,7,8,9,10]. These designations are based on spectral shifts of the absorption band in the aggregates compared to the monomer state; H-aggregate is characterized by a blue-shift and the J-aggregate by a red-shift [11,12]. Moreover, by controlling pathway complexity, different types of assemblies can be generated from a single monomer [8,9,10]. However, self-assembly of such molecules has been much less studied in aqueous solution where most biological processes occur, especially for porphyrins, due to the difficulty of modulating the assembly pathway by the methods developed in the organic phase [8,9,10,13].

Development of the self-assembling nanostructures induced by biologically important molecules through diverse pathways in aqueous media is needed [6,14] because of the biological significance of water as well as the tunability, selectivity, and adaptability of hydrophilic supramolecular structures [15]. Reversible self-assembly of water-soluble molecules induced by adenosine triphosphate (ATP) and phosphate-hydrolyzing enzyme in aqueous solution has recently been reported [15,16,17,18,19]. Recently, our group reported the ATP-induced assembly of helical nanofibers from a water-soluble cationic porphyrin that featured achiral ammonium side groups [20]. A second study demonstrates that the structure of the aggregates depends on the substitution pattern at the porphyrin meso positions [21]. Reversible disassembly of the nanofibers could be achieved by alkaline phosphatase via hydrolyzing ATP [20]. The nanofibers were also demonstrated to be able carry out in-vivo delivery of the porphyrin photosensitizer to a tumor site for efficient photodynamic therapy of cancer [22]. The current study builds on these earlier results by exploring the relationship between the porphyrin side chain structure and the resulting aggregates (H- or J-aggregate structure) that are formed in the presence of pyrophosphate (PPi) and adenosine triphosphate (ATP). In other previous studies of cooperative interactions between branched poly(ammonium) side groups of conjugated polyelectrolytes and anions, we found that the concentration of aggregation-inducing polyphosphates needed to induce aggregation could be decreased by several orders of magnitude compared to monophosphate [23,24]. 

Within the context of these previous studies, in this work we demonstrate chiral self-assembly of different cationic side chain modified porphyrins (**1** and **2**, Figure 1a) through different pathways which are dominated by π-π interactions and electrostatic interactions between water-soluble cationic porphyrins and the polyphosphate ions. We found that by modifying the side chains of the porphyrin macrocycle it is possible to dramatically influence the nature of the aggregate structures induced by polyphosphate ions. In addition, it is shown that ALP catalyzed hydrolysis of the polyphosphate ions destabilizes the assemblies, resulting in reversible disassembly of the chiral aggregates, accompanied by loss of the induced circular dichroism (Figure 1b).

## 2. Results

*Absorption and Emission Spectroscopy.* Porphyrins **1** and **2** (Figure 1a) were synthesized and fully characterized according to the procedures detailed in the Appendix A. The two porphyrins each feature ammonium side chains linked to the tetraphenylporphyrin macrocycle via chiral L-alanine units (Figure 1a). Porphyrin **1** features linear mono-ammonium side chains, while porphyrin **2** is functionalized with branched side chains that feature three ammonium units on each side chain. Importantly, both porphyrins **1** and **2** are chiral and optically active due to the presence of the L-alanine units.

UV−visible absorption and fluorescence spectroscopy were conducted to determine the type of porphyrin aggregates formed in water. Aggregation typically causes self-quenching of the porphyrin fluorescence, as well as a decrease in oscillator strength and shift in Soret absorption band [3,7,8,9,10,25]. We found that the addition of PPi or ATP decreased the absorbance of **1**, but without inducing a significant shift of the Soret band maximum wavelength (Appendix A). Quenching of the fluorescence of **1** by addition of PPi or ATP was also observed (Appendix A), showing that PPi is a more efficient quencher than ATP. This is consistent with the absorption spectra in which it was seen that addition of PPi induced a larger decrease in the absorptivity of **1** than ATP does. 

Addition of PPi to **1** also gave rise to quenching of the fluorescence lifetimes (Appendix A, top). The decay trace for **1** was fitted by monoexponential function, while with addition of PPi the decays are fitted by biexponential function (Appendix A), which suggests the presence of free and aggregated **1**. On the other hand, the addition of ATP leads to a slight increase in the fluorescence lifetimes (Appendix A and Appendix A). The difference could be explained by considering the structural difference between PPi and ATP. In particular, PPi interacts only with the positively charged ammonium side chains and forces the porphyrin rings to form a “tight” aggregate structure in which the porphyrin rings are in close proximity. By contrast, ATP is larger and can interact, possibly via the aromatic base intercalating in between the porphyrin rings, preventing the porphyrin units from coming into close proximity in the aggregate structure.

For porphyrin **2**, we observed very different and interesting results that are induced by the addition of PPi or ATP. In the Soret absorption of **2**, addition of PPi caused a distinct blue shift of ~15 nm, whereas addition of ATP induced a clear red shift of ~8 nm. In both cases, a significant decrease in the absorptivity (Figure 2a,c) was observed. These spectral changes suggest that H-aggregates (blue shift) of the porphyrin are formed by face-to-face stacking induced by addition of PPi, whereas a J-aggregate (red shift) is produced by addition of ATP through side-by-side stacking of the porphyrin rings. The different types of aggregation that occur is probably due to the steric effect of ATP and its aromatic base, which could interact with the porphyrin rings to prevent them from forming the face-to-face H-type aggregates (Figure 1b). In contrast, PPi can easily fit into the space between two adjacent branched side chains of **2** and facilitates porphyrin molecules to adopt a tightly packed and twisted H-aggregate structure. 

Different trends were observed in the fluorescence of **2** upon addition of PPi and ATP. The fluorescence of **2** was strongly quenched when the concentration of PPi increased from 0 μM to 30 μM, but additional quenching was not observed when the PPi concentration was greater than 30 μM (Figure 2d), indicating that above 30 μM, all of the porphrin was aggregated. Analogous studies by time-resolved fluorescence spectroscopy carried out on the emission of **2** with addition of PPi clearly demonstrated the interactions between **2** and PPi. Concomitant with the quenching in fluorescence intensity of **2**, the fluorescence lifetime becomes shorter with the addition of PPi (Appendix A, bottom and Appendix A). This behavior indicates that when the PPi concentration is higher than 30 μM, porphyrin **2** aggregates are the dominant species, in good agreement with the steady-state fluorescence titration experiment where the quenching is saturated at ~30 μM PPi concentration (Figure 2d).

Interestingly, the addition of a low concentration of ATP gave rise to an initial increase in fluorescence intensity and fluorescence lifetime; further increase causes quenching, albeit less strongly than caused by PPi (Figure 2b and Appendix A). Concomitantly, the addition of ATP to **2** caused a decrease in the Soret band absorptivity and a red shift (Figure 2a). Taken together, these spectroscopic results point to the fact that the structure of the aggregates formed by mixing **2** with PPi and ATP are structurally distinct, suggesting an H-type aggregate predominates with PPi and a J-type aggregate with ATP. Comparison of the Stern–Volmer plots for fluorescence quenching of **1** and **2** indicates that the quenching is efficient, with K_SV_~10^4^ M^−1^ for ATP and K_SV_~10^5^ M^−1^ for PPi (Appendix A). For an aggregation-caused-quenching system, such high quenching efficiencies are usually only observed in polymeric chromophore systems where the quenching is amplified [23,24], but not in small molecule-based chromophores. The relatively high Stern–Volmer quenching constants that are seen for PPi and ATP quenching of **1** and **2** are consistent with a static quenching mechanism, where quenching is caused by the phosphate–induced aggregation which is likely a cooperative process, taking place at relatively low concentration of the added phosphate ions.

The absorption spectra of titration experiments of **2** by nucleoside analogs of ATP such as cytidine triphosphate (CTP), guanosine triphosphate (GTP) and uridine triphosphate (UTP) all showed red-shifted Soret bands (Appendix A), but the shifts were not as distinct as when ATP is added (Figure 2a), presumably due to weaker interactions with porphyrin compared to ATP. Nevertheless, this result still indicates that nucleoside triphosphates (NTPs) interact with **2** in a similar way to form J-aggregates. In the case of the addition of triphosphoric acid (TriPi, Appendix A), which is structurally-analogous to PPi, we observed distinct blue-shifted Soret bands in the absorption spectrum of **2**, indicating H-aggregate formation.

*Molecular Dynamics Simulations.* To provide more insight into the supramolecular structure of **2** induced by PPi and ATP, molecular dynamics (MD) simulations were employed using a methodology adapted from previous work [20], as described in the Appendix A. The final structures that result from the MD simulations are shown in Figure 3, with explicit water molecules hidden for clarity. These simulations used periodic cell conditions so that the structures repeat for every four porphyrins. Because a Parrinello–Rahman barostat was used, cell angles equilibrated over the course of the MD run, resulting in final cell angles of roughly 90° each (+/− 3°) for simulations with PPi, while the simulation cell containing ATP became distorted with the cell angle corresponding to that with respect to the plane of the porphyrin rings reaching 61° after final relaxation. The simulation results corroborate the spectroscopic findings where PPi induces H-aggregates and ATP induces J-aggregates. In particular, for the PPi aggregate, the porphyrin rings are aligned (like a stack of cards), but they are angled and offset (slipped stack) in the ATP aggregate. The spacing between porphyrin planes was measured as 4.3 Å (typical for π-π stacking interactions) for the PPi-induced aggregate, and 6.6 Å within the ATP aggregate. The larger spacing between rings within the ATP aggregate is explained by the intercalation of the aromatic base of ATP into the porphyrin stack (Figure 3, top right), whereas the PPi ions are clustered around the charged side chains of **2** (Figure 3, top left). The number of ATP molecules intercalated between two porphyrins in the supramolecular structure varied between one and three, limited by sterics, in which the adenosine base formed an intercalated layer that separates adjacent porphyrins from one another. The intercalated aromatic base of ATP in conjunction with the long tri-phosphate chains interacting with the branched ammonium groups allowed for the porphyrin rings to slide into a J-aggregate structure, where the rings are tilted and offset, and also increased the distance between the porphyrins in the aggregate structure.

*Circular Dichroism Spectroscopy*. Given that the **1** and **2** are chiral due to the presence of the L-alanine stereocenters in the porphyrin side chains, we anticipated that the aggregates may exhibit circular dichroism (CD) due to the chiral induction by the stereocenters. Importantly, solutions of **1** and **2** alone exhibit little or no CD signals (Figure 4a and Appendix A). This is not surprising, given that the L-alanine stereocenters are spatially isolated from the porphyrin macrocycle which is the basis for the chromophore. By contrast, aggregates of **2** induced by PPi or ATP exhibit distinct CD signals in the region of the Soret absorption band (Figure 4). The CD spectra of **2-**PPi and **2-**TriPi revealed that PPi or TriPi both induced formation of an H-aggregate with a right-handed helical sense, as indicated by the bisignate CD couplets with a positive Cotton effect [10]. Even though the CD signal of assemblies of **1** with PPi (**1-**PPi) shows a bisignate feature (Appendix A), the amplitude of the CD signal is much lower than that of **2-**PPi, indicating that the monomer units in **1-**PPi are less ordered or are weakly coupled, in accordance with the UV-visible absorption spectra which exhibited no obvious band shift. The addition of the various nucleoside triphosphates (NTPs) to **2** induced only a small CD signal or non-bisignate CD bands peaked at the Soret band maximum in their respective UV-visible absorption spectra, confirming that the aggregates are not helical or H-type (Figure 4b). The weak CD signal from **2-**NTPs could be attributed to the chiral amplification effect from the aggregates [26,27]. Based on the UV-visible absorption and CD spectral shape of the **2**-polyphosphate complexes (Figure 2 and Figure 3 and Appendix A), we can easily distinguish PPi or TriPi from the other polyphosphates such as the NTPs. It is because of that only PPi or TriPi induce a blue-shift to Soret band of porphyrin **2** and at the same time give rise to right-handed bisignate CD bands. Adding ATP to **1** causes no CD signal change of **1**, which displays a very weak CD signal (Appendix A), indicating that no ordered structure or large aggregates form, which is also consistent with the absorption spectrum showing only a slight change upon addition of ATP.

*Aggregate Size and Structure.* Dynamic light scattering (DLS) analysis of aqueous solutions of **1** and **2** (Appendix A and Figure 4a) revealed that they exist as monomers (*d* < 1 nm) in water (pH 6) [7]. This is distinct from the reported self-assembling systems that are based on ionic naphthalene diimides which are self-aggregated in water [15,16,17,18]. DLS revealed a large size increase to ~100–200 nm when PPi was added to solutions of **1** or **2**. However, when ATP was added, a large increase in size to ~200 nm was observed for **2**, but for **1** the size increase was much smaller (~5 nm, Appendix A). These results indicate that PPi causes **1** and **2** to form relatively large aggregates, but ATP only induces the formation of large aggregates of **2**.

Transmission electron microscopy (TEM) and selected area electron diffraction (SAED) measurements (Appendix A) confirmed that ATP induced the formation of relatively small, semi-crystalline nanoparticles of **1** with diameters of~5 nm. The addition of PPi to **1** yielded considerably larger, vesicle-like structures which appear to be hollow (Appendix A). In contrast, the addition of ATP to **2** led to the formation of nanoparticles with diameters of ~100 nm, as confirmed by DLS (Figure 5a) and TEM (Figure 6b). The TEM image of **2** with 3.0 equivalents of PPi shows that the assemblies are plate shaped (Figure 6a), possibly due to the formation of supramolecular bundles by porphyrin-based H-aggregates linked by PPi.

*Reversible Aggregation-Deaggregation Induced by ALP*. Previously, our group reported the assembly and disassembly of a conjugated polyelectrolyte/PPi aggregate by using the enzyme alkaline phosphatase (ALP) to regulate this process [24]. The polyelectrolyte/PPi aggregate is disassembled during the ALP catalyzed hydrolysis of PPi to Pi. More recently, we also reported the use of ALP in catalyzing the disassembly of helical fibrous nanostructure of porphyrin-ATP co-assemblies that can be monitored by CD and DLS measurements [20]. As shown herein, porphyrin **2** also exhibits a high binding affinity towards PPi or ATP in water and forming H-aggregates or J-aggregates, respectively. To investigate the effect of ALP catalyzed hydrolysis on the **2**-PPi or **2**-ATP aggregates, ALP was added to the aggregates, and the mixed solutions were monitored by absorption spectroscopy and DLS. Interestingly, when ALP was added, the H-aggregates or J-aggregates were disassembled, releasing the free porphyrin monomers. When the ALP-catalyzed phosphate hydrolysis process was complete, a distinct color change from yellow (the complex) to pink (free porphyrin) was observed (Appendix A), and DLS data showed that the scattering size decreased to <1 nm (Figure 5b and Appendix A). Absorption spectra (Appendix A) obtained after the hydrolysis of **2**-PPi or **2**-ATP were almost the same as that of a pure solution of **2**, which also confirms that the H- or J- porphyrin aggregates were disassembled to monomer **2**. The slightly decreased extinction coefficient of the Soret band is attributed to the presence of the hydrolysis product phosphate ion (Pi), which causes a slight decrease in the extinction of the Soret absorbance (Appendix A).

CD spectra of **2-**PPi also showed a decreased amplitude after addition of ALP, and the bisignate couplet completely disappeared over time, leaving only the weak signal originated from the chiral monomers (Appendix A). DLS correlation function curves, which were collected for every 90 s (Appendix A) during the hydrolysis process of **2-**PPi or **2-**ATP by ALP, clearly indicate a gradual decrease in the size of the aggregates. More importantly, **2** could reassemble into J-aggregates or H-aggregates by subsequent addition of ATP or PPi, confirming that phosphate-induced self-assembly is reversible.

## 3. Discussion

The studies reported herein probed the structure of aggregates formed by the interaction of cationic porphyrins substituted with ammonium groups with polyphosphate ions in aqueous solution. In every case, we found that the interactions lead to the formation of supramolecular aggregates; however, the details of the aggregate structures depend on the porphyrin structure as well as on the polyphosphate ion. The two porphyrins studied herein differ in the number of ammonium side groups. Porphyrin **1** is substituted with four side groups that feature only one ammonium group, imbuing **1** with a (4+) charge. The side groups in porphyrin **2** are branched and each have three ammonium groups, and as a result, the overall charge on **2** is (12+). 

Both porphyrins interact relatively strongly with PPi, with the interaction being slightly stronger for **2**. Modeling suggests that the PPi ions interact with the ammonium groups and reside at the periphery of the aggregate structure, providing ionic links between adjacent porphyrin rings which are strongly interacting in a π-stacked array. The fluorescence quenching data are consistent with a strong interaction between the porphyrin rings in the PPi aggregates, as reflected by the strong steady state quenching and decreased lifetime. In contrast to PPi, ATP interacts more weakly with both porphyrins, and the nature of the aggregates formed is different compared to PPi. In the case of the **2**-ATP aggregate, the spectroscopy suggests a J-aggregate structure in which the porphyrin rings are offset along the long axis. In addition, the modeling suggests that the adenosine base is intercalated in between the porphyrin macrocycles and this leads to a decrease in the interaction of the porphyrin chromophores. This is consistent with the observation of weaker fluorescence quenching for the **2**-ATP aggregate.

## 4. Materials and Methods

Solvents were dried by standard methods or by elution through a MBruan MB-SPS-800 solvent purification system or were purchased as anhydrous. Palladium catalysts and copper(I) iodide were used as received from Sigma-Aldrich. All other chemicals were purchased from either Acros or Sigma-Aldrich and used without further treatment. The porphyrins were dissolved in deionized water as stock solutions and the concentrations were adjusted to 1.0 mM. The stock solution was diluted to the needed concentrations before the experiments. 

Reactions were monitored by silica gel TLC plates, visualized with ultraviolet light, or iodine stains. Column chromatography was performed on 200-400 mesh silica gel. 1H, 13C-NMR spectra were recorded on a 500 MHz Bruker Avans spectrometer using 1,1,1,1-tetramethylsilane (TMS) as internal standard. Water used for all studies was Milli-Q deionized water (18.2 MΩ.cm). High resolution mass spectrometry (HR-MS) was collected on a maXis plus quadrupole-time of flight mass spectrometer equipped with an electrospray ionization source (Bruker Daltonics) and operated in the positive ionization mode. TEM analyses were carried out on a JEOL 2010F (200 kV) HR-TEM. Samples were prepared by depositing 5 μL of the sample on a 400-mesh carbon-coated copper grid (Ted Pella, Inc.), which was treated with chloroform prior to use. Samples were allowed to adsorb on the grid for 5 min, and then excess sample was wicked with a piece of filter paper. Then the grid was dried under air. UV-visible absorption spectra were recorded on a Shimadzu UV-1800 dual beam spectrophotometer. Corrected steady-state emission spectra and time-dependent fluorescence intensities were measured on an Edinburgh FLS 1000 photoluminescence spectrometer. Dynamic light scattering (DLS) size measurements were recorded with a NanoBrook (BrookHaven US) employing a 640-nm laser at 90° relative to the incident beam or at a back-scattering angle of 173°. Circular Dichroism (CD) measurements were recorded on a Jasco J-815 spectrometer. The temperature was maintained using a CDF-426S/15 Peltier-type temperature controller. For all the measurements, the scan rate was chosen appropriately. A 10-mm quartz cuvette was used in all cases. All solutions for measurements of enzyme-mediated transient self-assembly were prepared in 50 mM MES solution (pH = 6.5), and a constant temperature of 38 °C was maintained throughout the measurements. Fluorescence lifetimes were measured in a PicoQuant FluoTime 300 Fluorescence Lifetime Spectrophotometer by time-correlated single photon counting (TCSPC) technique (PicoQuant PicoHarp 300 module). A Ti:Saphire Coherent Chameleon Ultra laser was used as excitation source. The Chameleon laser provides pulses of 80 MHz that can be tuned from 680-1080 nm. The 80 MHz pulse train was sampled using a Coherent Model 9200 pulse picker to provide excitation pulses at 4000 kHz repetition rate and was frequency doubled using an Coherent second harmonic generator. An excitation wavelength of 515 nm was chosen for all samples and the instrument response function (IRF) was measured by using a Ludox scattering solution.

Molecular mechanics simulations were carried out in the Materials Studio 2018 suite of software from Biovia. The methodology followed that of previous work [20]. A total of four porphyrins were placed into a periodic cell using the Amorphous Cell module, along with a charge-balancing quantity of PPi or ATP and sufficient water molecules to comprise 50% weight. The Compass II forcefield was used for all optimizations and dynamics calculations. The simulation was started at low density (0.1 g/mL) with atomic charges reduced to 10% of their forcefield assigned values to prevent the formation of metastable jammed states. Porphyrins with PPi or ATP spontaneously formed face-to-face stacks during 1 ns of NVT dynamics. After this a series of cell relaxations and NPT dynamics with a Berendsen barostat were used to guide the stacks together so that they interacted facially, as the "docking" mechanism of two stacks of porphyrins depends only on their relative positions at the time of cell relaxation. After this phase, the simulation cells reached an equilibrated density. To ensure that these interfacially-docked stacks are stable, simulation cells were subjected to 3 ns of NVT annealing from 300 K to 650 K with 1 GPa of external pressure applied. After this, atomic charges were restored to their forcefield-assigned values, temperature set to 298 K, and pressure lowered to 1 atm for a production run of NPT dynamics for 5 ns using the Parrinello-Rahman barostat. After this, final geometries were optimized and the results of Figure 3 were extracted from there. All dynamics calculations used a timestep of 1 fs. An Ewald summation method was used for electrostatic interactions and an atom-based approach was used for van der Waals interactions with a cutoff distance of 12.5 Å. The Nosé thermostat was used for monitoring temperature.

## 5. Conclusions

This work demonstrates the formation of different types of water-soluble-porphyrin-based supramolecular assemblies (H-type or J-type aggregation) induced by different phosphate anions, and the enhancement of the formed specific type of aggregation through side chain engineering. This study opens a new direction of utilizing electrostatic interaction to mediate the self-assembling outcomes by tuning the preferred self-assembly pathways via adding different inducers. Owing to the tunability and reversibility of this self-assembly system, it holds the potentials to achieve the construction of active functional materials with dynamic nature and adjustable structure, as well as to mimic the naturally-occurring biomaterials or develop novel drug delivery systems with controlled-release capability to adapt to the biological environment. 

## Figures and Tables

**Figure 1 molecules-28-04115-f001:**
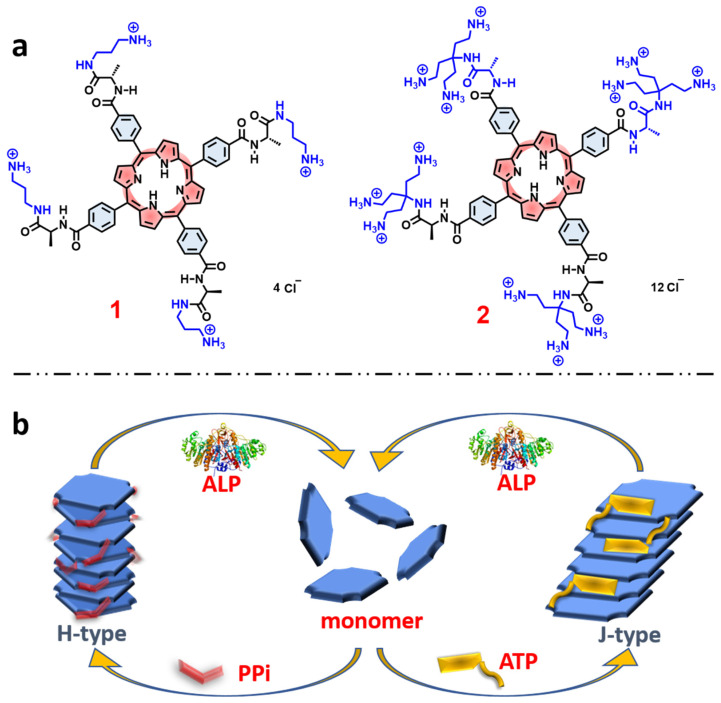
(**a**) Molecular structures of the porphyrin derivatives **1**, and **2**. (**b**) Schematic representation of self-assembly pathways of porphyrin monomers induced by different phosphates and disassembly by ALP.

**Figure 2 molecules-28-04115-f002:**
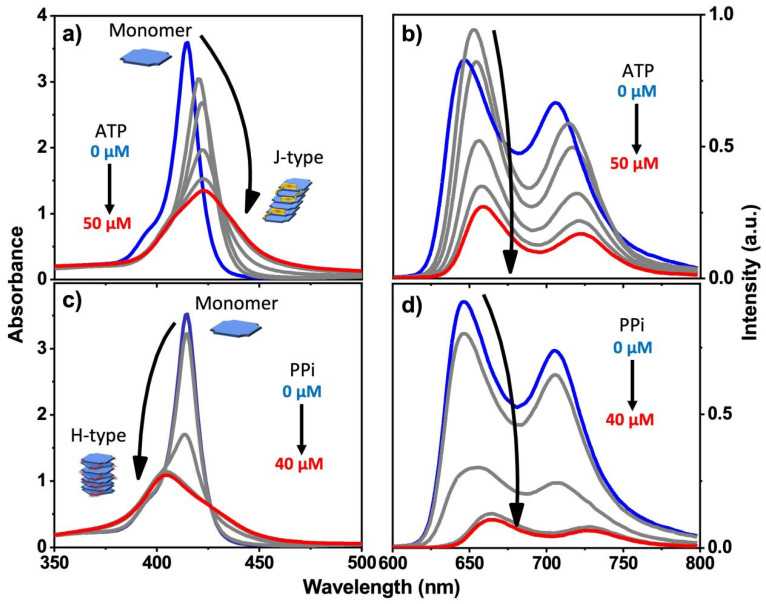
Absorption spectra (**a**,**c**) and fluorescence spectra (**b**,**d**) of **2** in water with increasing concentration of PPi or ATP. (pH = 6; [2]: 10 μM in all cases). Blue plot is 0 μM ATP or PPi, red plot is final 40 or 50 μM ATP or PPi.

**Figure 3 molecules-28-04115-f003:**
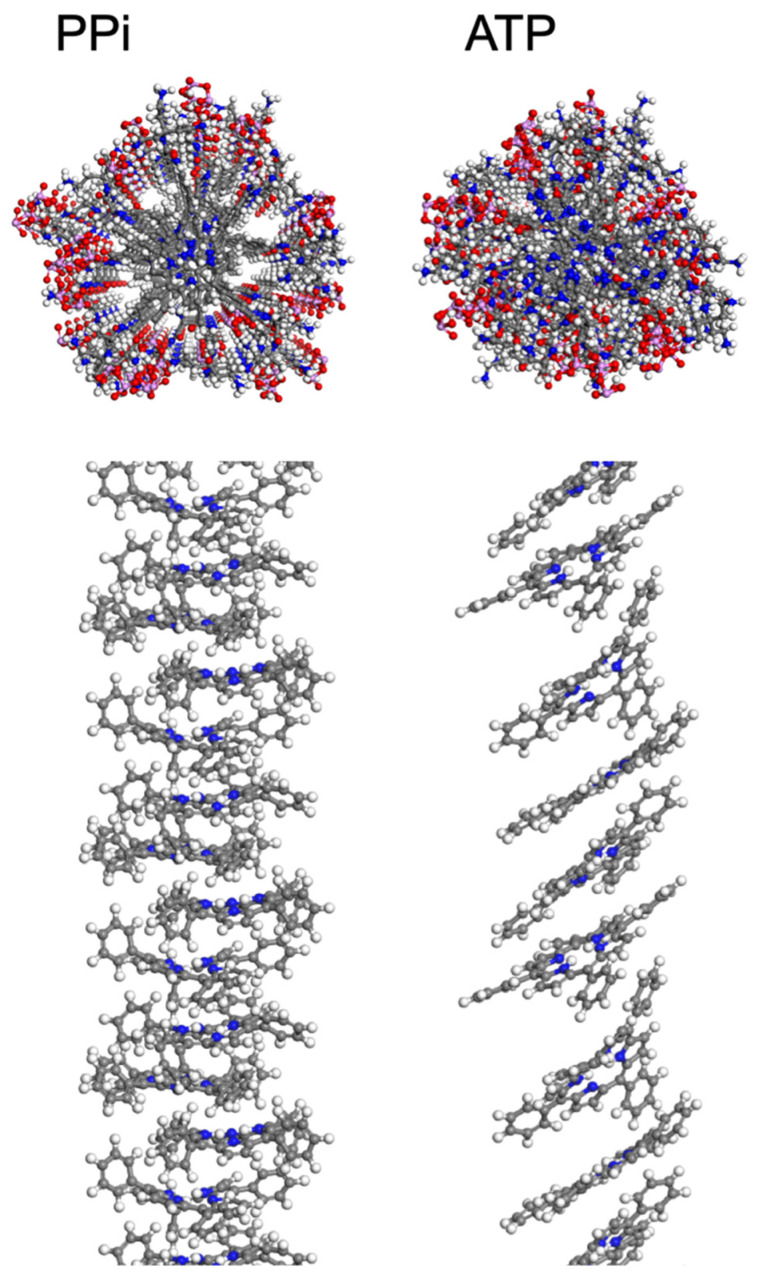
Structures resulting from MD simulations of **2** in the presence of PPi (**left**) and ATP (**right**) with explicit water molecules hidden. These snapshots were extracted from a periodic simulation cell containing four porphyrins per cell. In the upper panels, the view is along the long axis of the aggregate, and the structure shown includes the porphyrin macrocycle, side groups, and phosphates. In the lower panel, side views of the aggregate structures are shown with the alkyl-ammonium side chains and polyphosphates hidden for clarity. Atom color scheme: C-gray, H-white, N-blue, O-red, P-pink.

**Figure 4 molecules-28-04115-f004:**
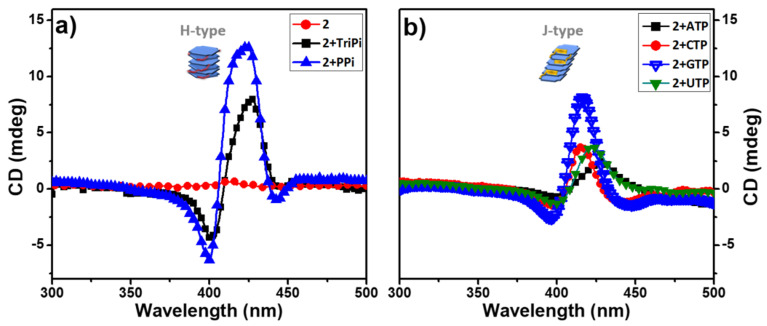
CD spectra of aqueous buffered solutions of **2** before and after adding 3.0 equivalent of PPi, TriPi or Nucleotide triphosphates (NTPs). (**a**) **2**, **2**+TriPi, **2**+PPi; (**b**) **2**+ATP, **2**+CTP, **2**+GTP, **2**+UTP. Conditions: [2]: 10 μM, [PPi] or [TriPi]: 30 μM, [NTPs]: 30 μM.

**Figure 5 molecules-28-04115-f005:**
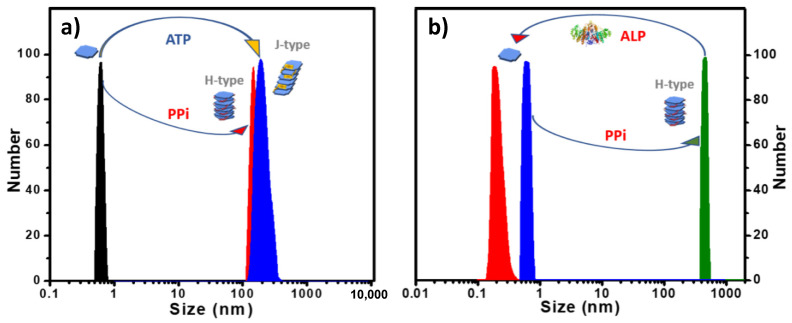
(**a**) DLS size distribution of **2** with 3.0 equivalents of PPi or ATP added in water. (**b**) DLS size distribution showing **2-**PPi assembly disassociation by treatment with ALP enzyme in MES buffer. ([2]: 10 μM in all cases).

**Figure 6 molecules-28-04115-f006:**
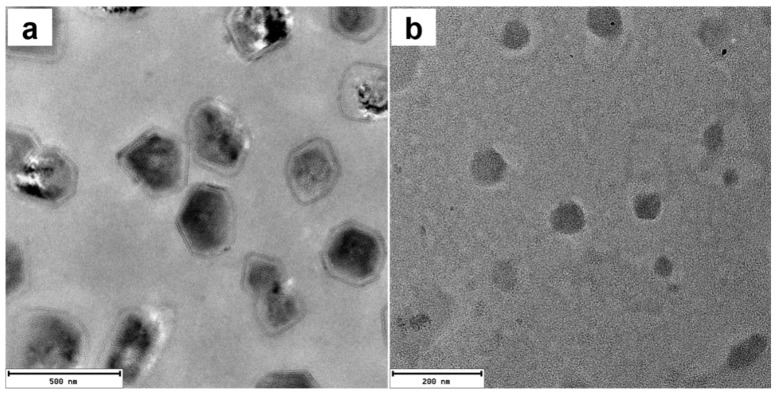
TEM images of **2** with 3.0 equivalents PPi (**a**) or ATP (**b**). Scale bar: (**a**) 200 nm; (**b**) 500 nm. ([2]: 10 μM).

## Data Availability

All the data generated by this research are included in the article.

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
