# Peer review of "Phosphates Induced H-Type or J-Type Aggregation of Cationic Porphyrins with Varied Side Chains"

_molecules, 2023, doi:10.3390/molecules28104115_

Round 1

Reviewer 1 Report

In this manuscript, two different types of side chain substituted chiral cationic porphyrins and their aqueous solution self-assembly were introduced. It not only shows that the interaction between cationic porphyrins and biological phosphate ions promotes more obvious H-or J-type aggregation, but also shows that phosphate-induced cationic porphyrin self-assembly is reversible in the presence of enzyme alkaline phosphatase and repeated addition of phosphate. In general, the manuscript is clear, logical, and reasonable, and should be accepted after minor modification.

1. Abbreviations are used instead of proper nouns and should not often defined, such as “PPi” refers to “pyrophosphate”, “ATP” refers to “adenosine triphosphate” and “CD” refers to “circular dichroism”, etc. The authors should check the whole context.

2. In support information, compounds 10 and 14 should be checked and corrected.

3. Lines 224-226, please revise the sentence for clarity.

Generally, it is good to understand.

Author Response

Please see attached word document

Reviewer 2 Report

The article "Phosphates Induced H-type or J-type Aggregation of Cationic 2 Porphyrins with Varied Side Chains" by Zhiliang Li and co-workers describes preparation and extensive spectroscopic studies of a pair of water-soluble porphyrins which form reversible aggregates in the presence of phosphate anions. The article is well written and is a pleasure to read. The observations denoted in the paper are important, since the aggregation of cationic porphyrinoids in anionic buffers is a common phenomenon, quite important for e.g. PDT and PACT, however not studied in good details.

I recommend to accept and publish the article essentially as is, with a few minor additions:

1) I would ask authors to stress that terms "J/H-aggregates" belong to the spectral behavior at the first place, i.e. to the bathochromic shift of absorbance and increased fluorescence for J and to the hypsochromic shift of absorbance and decreased fluorescence for H (see original JELLEY, E. E. (1936). Spectral Absorption and Fluorescence of Dyes in the Molecular State. Nature, 138(3502), 1009–1010. and e.g. Angew. Chem. Int. Ed.2011,50, 3376 – 3410)

2)  Black arrow on Figure 2b is not well visible, please redraw somehow

3) Please add brutto-formula to the MS data in supplementary information
